# Transfer Prediction Method of Bearing Remaining Useful Life Based on Deep Feature Evaluation under Different Working Conditions

**DOI:** 10.3390/s23198254

**Published:** 2023-10-05

**Authors:** Yongzhi Liu, Yisheng Zou, Kai Zhang

**Affiliations:** 1School of Mining and Mechanical Engineering, Liupanshui Normal University, Liupanshui 553004, China; 2School of Computing and Artificial Intelligence, Southwest Jiaotong University, Chengdu 610031, China; zysapple@swjtu.edu.cn; 3School of Mechanical Engineering, Southwest Jiaotong University, Chengdu 610031, China; zhangkai@swjtu.edu.cn

**Keywords:** bearing remaining useful life, transfer prediction, feature evaluation, different working condition

## Abstract

In the existing bearing remaining useful life (RUL)-prediction model based on deep learning, the advantages and disadvantages of the extracted features are evaluated by the prediction accuracy; thus, the analytical ability of the features is poor. At the same time, the change of working conditions has a great influence on prediction accuracy. To overcome these limitations, a prediction method of bearing RUL based on feature evaluation and deep transfer learning is proposed. The proposed model can solve the above problems: (1) a method of feature evaluation and selection for bearing life prediction based on trend consistency index was designed. (2) In this study, a domain adversarial transfer model based on feature condition mapping is proposed to overcome the second limitation. Experimental results show that this method is superior to the existing bearing evaluation and prediction methods.

## 1. Introduction

Rolling bearing is one of the key parts of mechanical equipment, and its reliability directly affects the operational safety of the equipment [1]. Thus, it is of great significance to predict the remaining useful life (RUL) of rolling bearing for the health evaluation of the entire equipment [2,3,4]. Rolling bearing RUL-prediction methods can be divided into model-based and data-driven methods [5,6]. The model-based method needs to make assumptions about the bearing degradation process, but it is quite different from the actual degradation process and requires artificial knowledge from experience, so the application is still limited. Currently, with the rapid development of intelligent technology and deep learning, data-driven methods have become a focus of research in academia and industry.

The data-driven bearing RUL prediction mainly includes two parts: feature extraction and prediction model construction. The features extracted by traditional data-driven bearing RUL methods are generally some statistical indicators, such as root mean square and kurtosis [7,8,9,10]. Machine learning models are widely used in prediction models, such as support vector machines [11], hidden Markov [12], Bayesian network [13], etc. However, these methods need to extract degradation features based on expert knowledge and experience and then select the appropriate machine learning model to predict based on the changing trend of features. Recently, with the development of deep learning technology and its strong nonlinear mapping learning ability, it has been introduced into rolling bearing RUL prediction. This method extracts features adaptively from the original signal and completes the prediction, which reduces the dependence of the intelligent prediction model on expert knowledge and avoids the workload of manual feature extraction. Ren [14] proposed a bearing RUL-prediction method based on a deep convolution neural network (CNN), which can significantly improve the prediction accuracy of bearings. Hu [15] proposed a bearing RUL model based on a deep belief network and diffusion process to improve the prediction accuracy and uncertainty. Chen [16] proposed a new depth convolution autoencoder based on quadratic function, which can generate the health index of bearing from the original vibration signal and can be better applied to RUL prediction. Pei [17] proposed an adaptive prediction method for bearing mass data and prediction uncertainty. The premise of the high prediction accuracy of the method based on deep learning is that there are enough data, and the training and test data come from the same distribution. If this condition is not satisfied, the performance of the deep learning method will decline or even fail. However, in practical applications, it is difficult to collect enough data with the same distribution, most of which are collected under different working conditions. Transfer learning is considered to be an effective way to solve such problems. Transfer learning is a learning process that uses the similarity between data or models to apply the knowledge learned in the source domain to the target domain to solve the problem of insufficient identically distributed data in the target domain. The current transfer prediction is to extract the same or similar features in the source and target domain through transfer learning to improve the prediction accuracy of the model. The key is to measure the similarity or difference between the source and target domain. Chen [18] proposed a kind of transfer convolution neural network to learn domain invariant features, using multi-core maximum mean to reduce the distribution difference and achieved good prediction results. Zhu [19] proposed a transfer learning method based on multilayer perceptron to solve the problem of distribution difference and improve the prediction results. Mao [20] proposed an RUL-prediction method based on deep feature representation and transfer learning and introduced a transfer learning algorithm to adjust the features of the target bearing and auxiliary bearing to improve the prediction accuracy.

Through transfer learning, the invariable or similar features in the source and target domain are extracted from the vibration signals. How to evaluate and screen these features quantitatively, reduce the information redundancy, and further make these features conducive to improve the prediction accuracy of the bearings RUL is a problem worth studying. Currently, the study of this kind of problem mainly includes a dimension reduction algorithm and evaluation index. In the aspect of dimension reduction algorithm research, there is principal component analysis [21], local preserving projection algorithm [22], and approximate diagonalization of eigenmatrix [23]. Zhang B [24] and others generated candidate prediction features by processing the time, frequency, and time–frequency domain of the original condition monitoring the signal and defined three indicators of time correlation, monotonicity, and robustness according to the trend and residual of the features.

Guo [25] extracted the time, frequency, and time–frequency domain features of bearing signal in the study of bearing RUL prediction and selected the most sensitive feature from the extracted feature set according to the monotonicity and time correlation measurement. The filtering threshold was proposed to select the best feature subset. Wang F [26] studied an RUL prediction method based on long short-term memory, which used time correlation, monotonicity, and robustness to comprehensively evaluate the advantages and disadvantages of features. Kang [27] proposed an adaptive method to determine the weight of each evaluation index when studying the RUL-prediction method of rolling bearing. Berlin [28] evaluated the time and frequency domain characteristics of bearing signals based on monotonicity and sensitivity evaluation. Gu et al. [29] used four evaluation indexes of time correlation, monotonicity, discreteness, and robustness to evaluate and screen the state characteristics of the engine. In the process of bearing life prediction, Liu et al. [30] collected the time domain, frequency domain, IMF component, Hilbert marginal spectrum feature, and entropy feature of bearing and comprehensively evaluated the feature using three indicators of time correlation, monotonicity, and robustness.

Through the summary and analysis of the above literature, it can be seen that a lot of study on the RUL of bearings based on deep learning has been performed at this stage. The related research of bearing RUL transfer prediction and characteristic evaluation has also been performed, but there are still the following limitations:(1)In the existing feature evaluation research of bearing life prediction, the time and frequency domain features of vibration signal are evaluated and screened, but the feature extracted from the deep learning model is not evaluated. The automatic feature extraction function of the deep learning model can reduce the complexity of manual feature extraction. Using feature evaluation methods to evaluate this kind of feature can reduce the influence of human factors and improve the interpretability of deep features. Thus, it is valuable to evaluate the features extracted from the deep learning model;(2)Currently, the premise for deep learning to obtain high prediction accuracy in bearing RUL is that there are enough data, and the training and test data come from the same or similar distribution. If these conditions are not met, the performance of the deep-learning-based prediction method will decline or even fail. However, in practical application, the distribution of training data and test set (prediction data) is often different due to the change of working conditions. The current transfer prediction method based on maximum mean discrepancy and other domain adaptation is to reduce the difference of the overall distribution of source and target domain data, which may lead to the extracted features’ lack of prediction resolution.

The contributions of this study are as follows:

Aiming at the first limitation, a feature evaluation and screening method for bearing life prediction based on a trend consistency index is proposed. The signal features extracted from the deep transfer learning model are evaluated and screened, and the screened features are used to predict the remaining service life of bearing. The effectiveness of the proposed method is verified by comparing it with the prediction results obtained by classical evaluation indexes (time correlation, monotonicity, and robustness).

Aiming at the second limitation, we propose a bearing RUL-prediction method based on feature evaluation and deep transfer learning. The framework of the method includes feature extraction and evaluation, prediction, and domain adaptation module. First, the unsupervised deep learning model convolutional auto encode coding network model was used to construct the feature extraction model to extract the source and target domain features. Second, the domain adaptation module based on domain antagonism was used to reduce the difference of feature extraction between the source and target domain, and the feature condition mapping learning mechanism was added to improve the prediction resolution. Then, a trend consistency index was added to evaluate the extracted features, and the features with high scores were extracted according to the index scores. Finally, the full convolution network model was constructed as the prediction model, and the filtering features were input for prediction. The superiority of the proposed method was verified by collecting data on bearing failures under different working conditions.

The remainder of the study is structured as follows. The proposed method is presented in Section 2. Experimental details, results, and analysis are stated in Section 3. Finally, conclusions are drawn in Section 4.

## 2. Proposed Method

A transfer prediction method of bearing RUL based on deep feature evaluation is herein proposed. The framework of the method includes feature extraction, domain adaptation, feature evaluation, and prediction module, as shown in Figure 1. The feature extraction module was constructed by an unsupervised convolutional autoencoder network model. The domain adaptation module consists of two layers of a fully connected neural network using Wasserstein distance to measure the difference between different distributions. In the feature evaluation module, the trend consistency index was used to select the features with high trend consistency to predict the bearing life. The prediction module is a three-layer convolutional network.

The feature extraction module adopts the unsupervised learning deep convolutional autoencoder network, and the network structure of the autoencoder is shown in Figure 2. The convolutional autoencoder network model is composed of a three-layer one-dimensional convolution layer and a three-layer one-dimensional deconvolution layer. The hidden layer features of the autoencoder network were pooled to provide features for the prediction and domain adaptation module. The parameters of the feature extraction module are shown in Table 1. The loss function of the network is the mean square error, and the specific formula is as given (Equation 1):(1)$$L_{p}=\frac{{\sum_{i=1}^{n}\left(y_{i}-{y_{i}}^{p}\right)}^{2}}{n}$$
where $y_{i}$ is the input value, and ${y_{i}}^{p}$ is the output value.

### 2.1. Domain Adaptation Module

The domain adaptation module uses the idea of a generative adversarial network (GAN) to solve the problem of measuring the difference between the source and target domain in domain adaptation. The source and target domain data with different working conditions were input into the generator at the same time. The generator was used for feature extraction, and then, the discriminator was trained through continuous adversarial. When the discriminator cannot determine the source of the distribution of the feature extracted by the generator, it can be considered that the feature extracted by the generator is no longer different from the feature extracted by the target domain. The discriminator consists of two layers of a fully connected neural network, and the number of neurons in the last layer is 1. To avoid the instability of the original GAN training, the discriminator module uses Wasserstein distance instead of Jensen–Shannon (JS) divergence to measure the difference between different distributions.

1.Wasserstein distance
(2)$$W\left(P_{r},P_{g}\right)=\underset{\left\|f_{w}\right\|}{\sup}E_{{x\sim P}_{r}} [f_{w}\left(x\right)]-E_{{x\sim P}_{g}} [f_{w}\left(x\right)]$$
where $P_{r}$ and $P_{g}$ is the probability distribution; $f_{w}\left(x\right)$ is the fitting function of Wasserstein distance. When $f_{w}$ satisfies the 1-Lipschitz constraint, the Wasserstein distance between distributions can be approximately evaluated by adjusting the parameters of $f_{w}$.

The Wasserstein distance between source domain feature distribution $P_{h^{s}}$ and target domain feature distribution $P_{h^{t}}$ is as follows:(3)$$L_{wd}(x^{s},x^{t})=\frac{1}{n^{s}}\sum_{x^{s}\in X^{s}}^{}D(f_{g}(x^{s}))-\frac{1}{n^{t}}\sum_{x^{t}\in X^{t}}^{}D(f_{g}(x^{t}))$$

2.Conditional mapping constraints
(4)$$f_{g}\left(x^{s}\right)=f_{g}^{’}\left(x^{s}\right)-f_{g}^{’}\left(x^{s}\right)⊙y$$
where $f_{g}^{’}\left(x^{s}\right)$ is the mapping function in source domain of the feature extraction module.y is the label. The ⊙ symbol represents the dot product operation symbol, where $x^{s}$ and ${\;x}^{t}$ are the data samples from the source domain and the target domain, respectively; D(x) is the mapping function of the domain discrimination module; and $f_{g}(x)$ is the mapping function of the feature extraction module.

To satisfy the 1-Lipschitz constraint, the gradient penalty $L_{grad}$ is imposed on $\theta_{d}$.
(5)$$L_{grad}\left(\hat{h}\right)={\left({\left\|\nabla_{\hat{h}}f_{d}\left(\hat{h}\right)\right\|}_{2}-1\right)}^{2}$$
where $\hat{h}=\varepsilon h^{s}+\left(1-\varepsilon \right)h^{t}$. ∇ is a gradient differential operator.

Wasserstein distance can be calculated approximately by the following formula.
(6)$$\underset{\theta_{d}}{L_{W}=\max}\left\{L_{wd}-{\gamma L}_{grad}\right\}$$
where γ is the penalty coefficient.

Thus, the objective function of the feature extraction model of transfer learning based on domain confrontation can be written as follows:(7)$$\underset{\theta_{g},\theta_{c}}{L_{O}=\min}\left\{L_{p}+\underset{\theta_{d}}{\max} [L_{wd}-{\gamma L}_{grad}]\right\}$$

### 2.2. Prediction Module

The prediction module is constructed by a full convolution neural network, which reduces the number of training parameters using the weight sharing and local connection characteristics of the convolution layer. Increasing the number of layers of the network can improve the nonlinear mapping ability. Thus, the multilayer CNN is used as the prediction model for experimental verification. The structure of the model is shown in Figure 3. The model has three network layers, and the last layer is the full connection layer, which is used to output the life prediction value $r_{h}$. Finally, the weighted smoothing method is used to smooth the prediction results.

The RUL percentage label y is used in model training ${\;y}_{h}$. It represents the percentage of RUL at the current time in the total life. The calculation formula is as follows:(8)$$y_{h}=\frac{L-h}{L-1}$$
where L represents the total number of times of data acquisition for the corresponding bearing; h represents the h-th data acquisition for the corresponding bearing.

### 2.3. Feature Evaluation Module

(1)Design idea of trend consistent life prediction evaluation index

Time correlation, monotonicity, and robustness play an active role in the quantitative evaluation and screening of bearing signal features. However, when the selected features are input into the bearing life prediction model, the consistency of the same feature in the trend of different bearings will have an impact on the bearing life prediction, and these three classic feature evaluation indicators are not the same. The consistency of this trend is not directly considered.

In the time direction, the same signal feature should be consistent in the trend between different bearings. The higher the degree of consistency, the higher the prediction result. Otherwise, the lower the prediction result. To evaluate the trend consistency and select the features with higher trend consistency to predict the bearing life and improve the prediction accuracy, a new calculation method was designed; that is, with the help of correlation calculation formula, the trend consistency of the same feature among different bearings was calculated, and the trend consistency life prediction evaluation index is thus proposed.

(2)Construction method of trend consistent life-prediction evaluation index

First, the extracted features were smoothed. Second, the features were compressed using normalization and down-sampling methods. Then, the correlation between features was calculated using the correlation calculation formula. Then, the mean value of a group of calculated correlation values was taken for processing. Finally, the score of the trend consistency index was obtained.

Suppose the r feature sequence of the i bearing is $X_{r}^{i}=\left(X_{r}^{i}(t_{1}),X_{r}^{i}(t_{2}),\ldots ,X_{r}^{i}{(t}_{k})\right)$. The trend term after exponential weighted moving smoothing is ${X_{T}}_{r}^{i}=\left({X_{T}}_{r}^{i}(t_{1}),{X_{T}}_{r}^{i}(t_{2}),\ldots ,{X_{T}}_{r}^{i}{(t}_{k})\right)$.

The main calculation process of the trend consistency index is as follows:

Step 1: The trend terms of the same feature series of different bearings are normalized (0–1) and dimensionally reduced. Without changing the trend of feature change, the same feature of different bearings has the same data length. The trend term ${X_{T}}_{r}^{i}$ of the feature sequence is $Z_{r}^{i}=\left(Z_{r}^{i}\left(1\right),Z_{r}^{i}\left(2\right),\ldots ,Z_{r}^{i}(s)\right)$ after normalization and down-sampling. The main calculation formula of down-sampling is as follows:(9)$$Z_{r}^{i}\left(f\right)=\left(\overset{t_{k}=R\left(\left(b+1\right)\times m\right)}{\underset{t_{k}=R\left(b\times m\right)}{\sum }}X_{T}{}_{r}^{i}(t_{k})\right)÷G\left(f\right)$$
(10)$$G\left(f\right)=\left(R\left(\left(f+1\right)\times m\right)-R\left(f\times m\right)+1\right)$$
(11)m=K/s
where s means that the time length of ${X_{T}}_{r}^{i}$ is divided into s intervals averagely, which is also the total number of features after down-sampling; *M* is the length of each interval; R (·) is an upward rounding function; G (f) is the length of the f interval; X is the value of the feature after the f-th interval down-sampling.

Step 2: The correlation Formula (9) is used to calculate the correlation between the same feature series of different bearings. The calculation results of the correlation value $q_{r}^{ij}$ of the same feature sequence between two bearings are arranged as follows:(12)$$q_{r}=\left(\begin{array}{cccc} q_{r}^{12} & q_{r}^{13} & \cdots & q_{r}^{1n}\\ & q_{r}^{22} & \cdots & q_{r}^{2n}\\ & & \ddots & \vdots \\ & & & q_{r}^{\left(n-1\right)n} \end{array}\right)$$
where n is the total number of bearings.

$q_{r}^{12}$ represents the correlation value between the first bearing’s r feature sequence $Z_{r}^{1}$ and the second bearing’s R feature sequence $Z_{r}^{2}$.

Step 3: The results calculated in step 2 are processed by means, and the score q is obtained. The calculation process is shown in the following formula:(13)$$Q_{r}=\left(\overset{i=\left(n-1\right),j=n}{\underset{i=1,j=2}{\sum }}q_{r}^{\mathrm{ij}}\right)÷E\left(i<j\right)$$
where E is the total number of $q_{r}^{ij}$ in $q_{r}$.

The calculated mean value meets the requirements of the range from 0 to 1, so the final score of the trend consistency index of the r-th feature sequence is Q, and the closer q is to 1, the more consistent the trend performance of the same feature on different bearings. The prediction of residual life is helpful to obtain better prediction accuracy using the feature of high consistency score of this trend.

### 2.4. Model Training

The verification process of the bearing RUL-prediction feature selection method based on the trend consistency index mainly includes the following four steps:

Step 1: Data preprocessing. First, the original vibration signal at each time point is transformed into a frequency domain signal by fast Fourier transform, and the training and test set are divided according to certain rules.

Step 2: Feature extraction. The unsupervised convolutional autoencoder network model is used to construct the feature extraction model, and the GAN-based domain adaptive module is used to reduce the distribution difference between the source and target domain data to train the feature extraction network. After the training, the training and test set are input into the network simultaneously to obtain the corresponding feature set.

Step 3: Evaluation and screening features. According to the index evaluation steps, the trend consistency index is used to evaluate the features of the training set; according to the score of the index, the high-quality features of the trend consistency index are extracted.

Step 4: Predict bearing RUL. The multilayer full convolution network model is used as the prediction model, and the high-quality feature set of the training set is used to train the network; after the training, the high-quality feature set of the test set is input to obtain the prediction value of the model.

## 3. Experimental Verification

The experimental results and analysis are divided into five parts. First, the experimental data are explained. Second, the model feature extraction is tested and analyzed. Third, the influence of domain adaptation on feature distribution is analyzed. Fourth, the trend consistency index of feature evaluation is calculated and analyzed. Fifth, the influence of index scores on the prediction results is analyzed. Finally, the prediction of bearing RUL based on different evaluation methods is compared and analyzed.

### 3.1. Experimental Data

The experimental data are vibration acceleration data collected from the accelerated life bench test of rolling bearing, which comes from the PHM data challenge [31] held by the Institute of Electrical and Electronics Engineers (IEEE) in 2012. The accelerated life test of rolling bearing is shown in Figure 4. The data set contains the full life-cycle vibration data of 17 rolling bearings under three working conditions, including 7 bearings under the first and second working conditions and 3 bearings under the third working condition, as shown in Table 2. The data acquisition method was used to collect 2560 vibration accelerations every 10 s until the vibration acceleration in the data description reaches the set threshold, and the bearing failure condition is stopped. In this study, bearing dataset 1 is the source domain, and bearing dataset 2 and 3 are target domain, respectively.

### 3.2. Feature Extraction

In the process of experimental verification, any bearing is selected from the bearing dataset 2 target domain data set as the test set, and the other six bearings are selected as the training set. Taking bearing2_3 2–3 as an example, the effectiveness of the method is illustrated. The original vibration and frequency domain signal of the first 0.1 s sample of bearing 2_3 are shown in Figure 5 and Figure 6, respectively.

After the training of the feature extraction model, 320 feature sequences of bearing 2_3 were obtained. Two feature sequences were randomly selected to show the effect of feature extraction. The first and 320 feature sequences selected are shown in Figure 7 and Figure 8.

It can be seen from Figure 8 that the feature extraction model can extract different features from the test set, indicating that the extracted features are diverse.

### 3.3. Domain Adaptation

It can be seen from Figure 9 and Figure 10 that the probability density distribution of data extraction features in the source and target domain is quite different before transfer learning. After transfer learning, the consistency of probability density distribution of features was significantly improved, and the difference was reduced. This shows that the feature extracted by transfer learning is insensitive to the change of working conditions, which is conducive to the RUL-prediction model to achieve higher prediction accuracy.

### 3.4. Calculation and Analysis of Trend Consistency Index in Feature Evaluation

In the feature evaluation, the extracted features of the training set were evaluated, and due to the length of limitation, the calculation process of bearing 1-1 and bearing 2_1 in the training set is shown in the form of pictures. The trend terms of the first feature sequence of bearings 1_1 and 2_2 are shown in Figure 11a and Figure 11c, respectively. After normalization and down-sampling in step 1, the feature sequences of the three bearings are shown in Figure 11b and Figure 11d, respectively. From the comparison of the two results before and after normalization and down-sampling, it can be seen that this step does not change the changing trend of the feature sequence and achieves the purpose of the first step. After the correlation calculation in the second step, the correlation values of all bearing features are arranged as shown in Table 3.

According to the results obtained in step 2, we calculated the data in Table 3 using the calculation method in step 3. Finally, the trend consistency index score of the first feature sequence was obtained, and the score is 0.683.

Each bearing feature set has 256 feature sequences. According to the above process, 256 score values were obtained, as shown in Figure 12.

It can be seen from Figure 12 that the score of the 23rd feature sequence is the highest, while the score of the 22nd feature sequence is the lowest. To verify whether the score is reasonable, the trend items of the 22nd, and 23rd characteristic series of bearing 1_1 and 2_1 after down-sampling are listed, respectively, as shown in Figure 13 and Figure 14. By comparing Figure 13 and Figure 14, we can see that the more similar the trend is, the higher the index score is, while the more different the trend is, the lower the index score is. The above phenomenon shows that the calculation method of the trend consistency index is reasonable.

The 256 scores calculated from 256 feature sequences were normalized, and the calculation results are shown in Figure 15. The threshold value was selected as 0.5, and the feature with a normalized score above 0.5 was retained.

According to the sequence number obtained above, the features corresponding to the sequence number were extracted in the feature set of the training and test set, which are the high-quality features considered by the trend consistency index.

From the above calculation process, it can be seen that the trend consistency index score of the feature series is relative, and the feature series with a high score is more consistent in different bearings than the feature series with a low score.

### 3.5. Influence of Index Score on Prediction Results

To explain the influence of feature scores on the prediction results, high-quality feature sets with scores above 0.5 and common feature sets with scores below 0.5 were used for training and prediction, respectively. Taking bearing 2_2 and 3_3 as test sets randomly, the prediction results corresponding to their high-quality and common feature sets are shown in Figure 16 and Figure 17, respectively.

From the results of Figure 16 and Figure 17, it can be seen that the prediction effect of the feature with a high score of trend consistency index is better than that of the feature with a low score, which shows that the more consistent the trend performance of the same feature on different bearings is, the better the prediction effect that will be obtained.

### 3.6. Comparative Analysis of Bearing RUL Prediction Results

Three typical and trend consistency indicators were used to evaluate and screen features, respectively. The high-quality feature sets obtained from each evaluation were used for training and prediction, and four groups of prediction results were obtained. To better illustrate the prediction effect of this method, 256 feature sequences were used for training and prediction, and a group of prediction results using full feature sequences was obtained.

The mean absolute error was used as the evaluation standard of prediction error, and the calculation formula is shown in Formula (Equation 14).
(14)$$\mathrm{MAE}=\frac{1}{H}\overset{H}{\underset{h=1}{\sum }}\left|\left(y_{h}-r_{h}\right)\right|$$
where *H* is the total number of predicted values of corresponding bearings.

Taking bearings 1_1~1_7 as the source domain and the six bearings in bearings 2_1~2_7 as the target domain, one bearing was selected as the test set in turn. According to this division method, the mean error of prediction is shown in Table 4.

From the statistical results in Table 4, it can be seen that the prediction effect of the trend consistency index is better than the other three indexes in the test sample, which shows that the trend consistency index can effectively select the high-quality feature set, which is conducive to reducing the prediction error from the feature set extracted by the deep learning model. Compared with the other prediction results, the comprehensive average of the errors is reduced by 20.5%, 21%, 20.3%, and 22.8%, which shows that the feature evaluation method is suitable for the signal features extracted by the deep learning model and can improve the interpretability of such features to a certain extent.

To show the prediction effect more intuitively and comprehensively, bearing 2-2 and bearing 3-3 were taken as examples to show the prediction results when they are used as test sets, respectively. The prediction results are shown in Table 5. It can be seen that the prediction accuracy of the trend consistency index is higher than that of the full feature series.

## 4. Conclusions

To address the problem of feature extraction and evaluation in bearing life prediction, a method of bearing RUL transfer prediction based on deep feature evaluation is proposed, which adopts a feature evaluation and screening method based on the trend consistency index. Finally, the bearing RUL can be predicted. The conclusions are as follows:(1)A feature extraction method based on transfer learning is proposed. The feature extracted by transfer learning is helpful to reduce the distribution difference of bearings under different working conditions and to build a prediction model to improve the prediction accuracy. The overall prediction accuracy is improved by 3.1%;(2)The trend consistency index is proposed to evaluate and screen the signal feature. Compared with the three classical indexes, the comprehensive average error obtained by this index is reduced by 21%, 20.3%, and 22.8%, respectively, which shows that this index can effectively screen out the signal features that are conducive to reducing the prediction error of bearing residual service life and provides new ideas for the signal feature evaluation method;(3)When the signal feature evaluation method is applied to the signal feature extracted by the depth transfer model, the comprehensive average error of the index prediction is reduced by 20.5% compared to the case without feature evaluation, which shows the applicability of the method to such features and can increase the interpretability of such features to a certain extent.

For transfer prediction, different evaluation indexes are used to achieve better prediction results on some bearings, and the trend consistency index is integrated with other indexes to further improve the prediction accuracy.

## Figures and Tables

**Figure 1 sensors-23-08254-f001:**
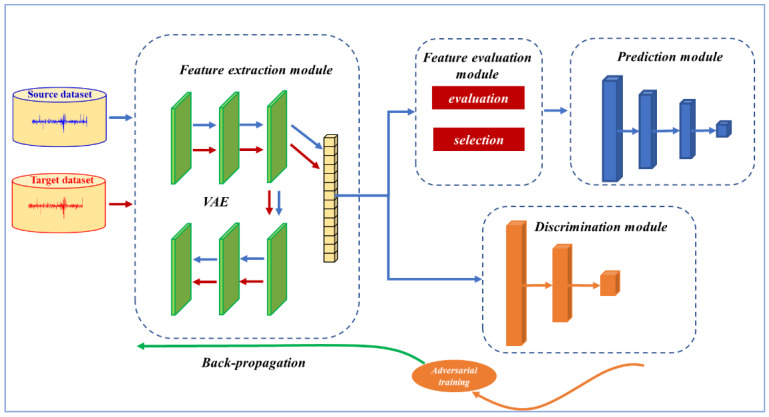
Feature Extraction Module.

**Figure 2 sensors-23-08254-f002:**
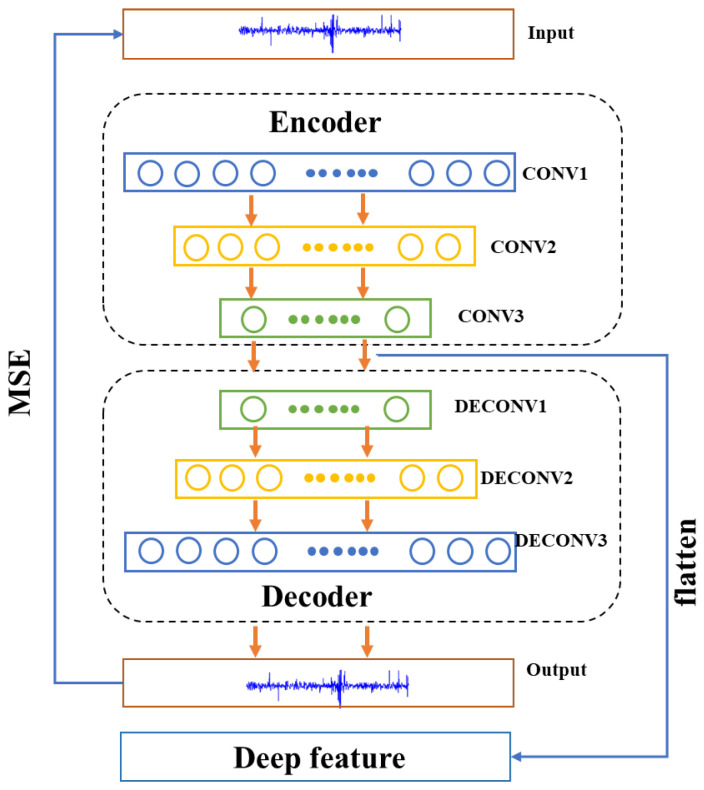
Feature extraction module.

**Figure 3 sensors-23-08254-f003:**
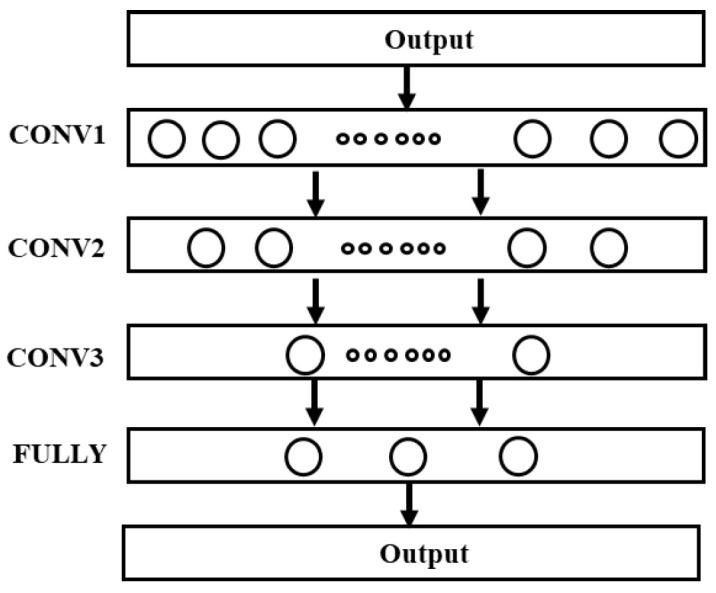
Network model for life prediction.

**Figure 4 sensors-23-08254-f004:**
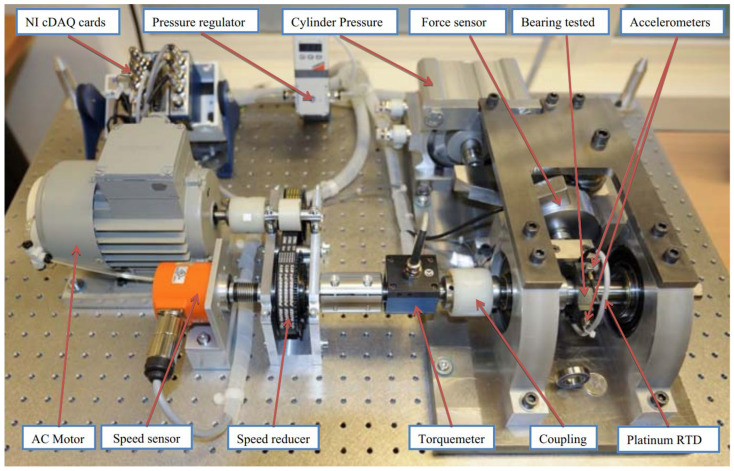
Accelerated life testbed of rolling bearing.

**Figure 5 sensors-23-08254-f005:**
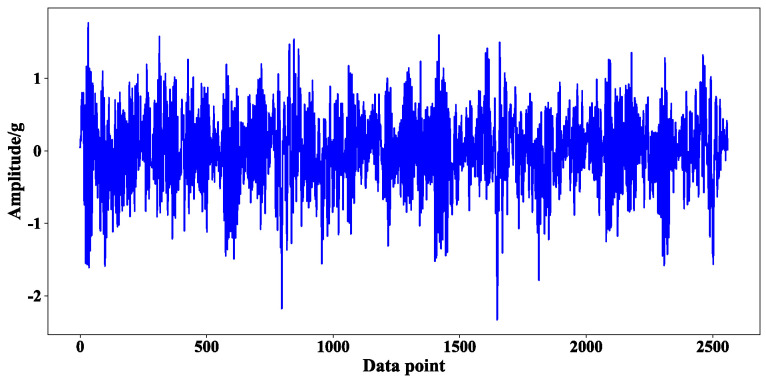
Original signal of the first 0.1 s sample of bearing 2_3.

**Figure 6 sensors-23-08254-f006:**
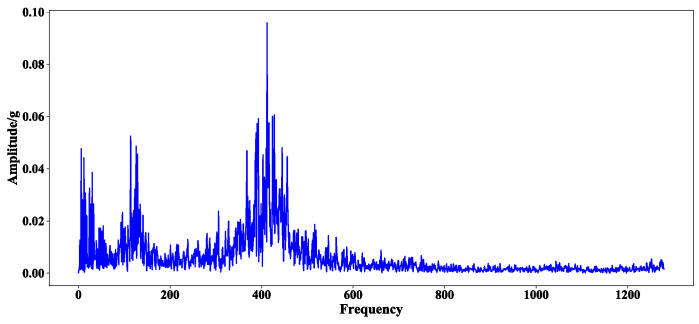
Frequency domain signal of the first 0.1 s sample of bearing 2_3.

**Figure 7 sensors-23-08254-f007:**
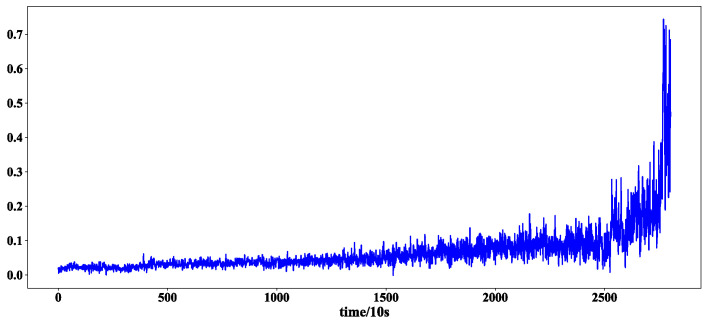
The first characteristic sequence of bearings 2-3.

**Figure 8 sensors-23-08254-f008:**
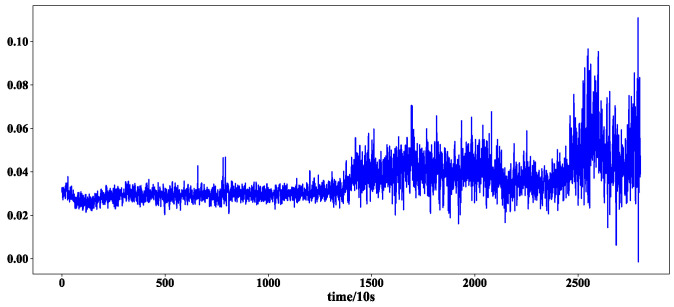
The 320th feature sequence of bearing 2-3.

**Figure 9 sensors-23-08254-f009:**
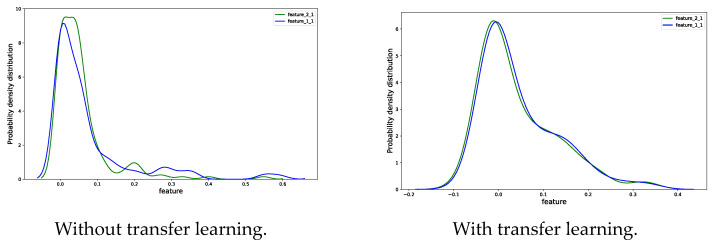
Feature probability density distribution of bearing 2_1 and bearing 1_1.

**Figure 10 sensors-23-08254-f010:**
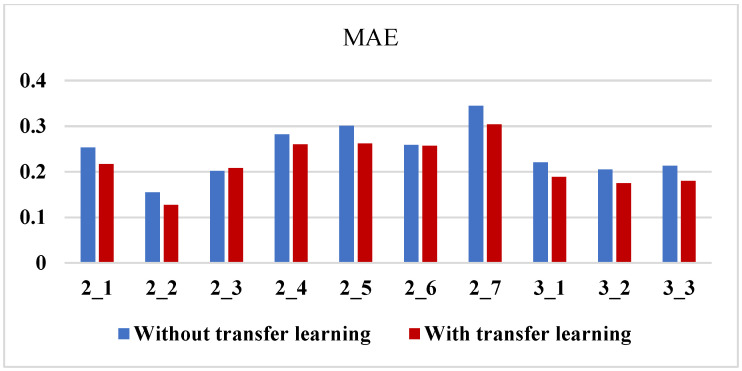
Feature of bearing without transfer learning and with transfer learning.

**Figure 11 sensors-23-08254-f011:**
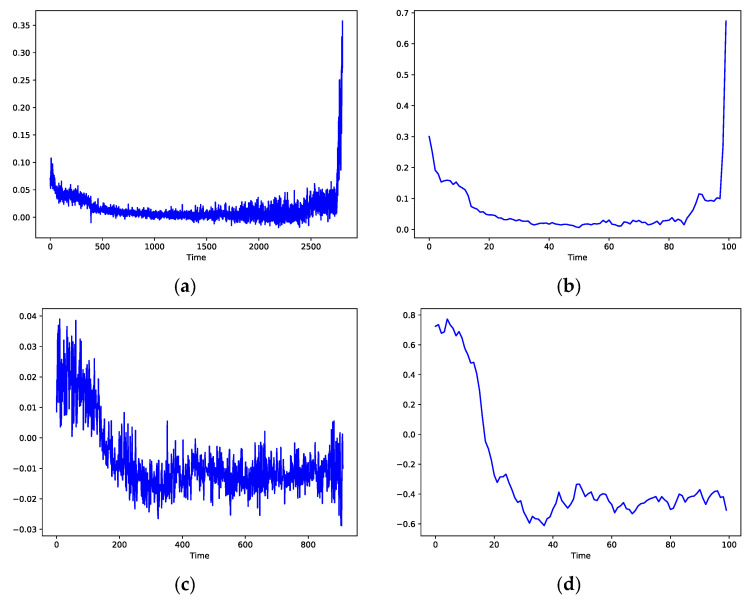
Trend items bearing 3_2 characteristics. (**a**) Trend term of bearing 1_1 feature. (**b**) Trend term of bearing 1_1 feature after down-sampling. (**c**) Trend term of bearing 2_1 feature. (**d**) Trend term of bearing 2_1 feature after down-sampling.

**Figure 12 sensors-23-08254-f012:**
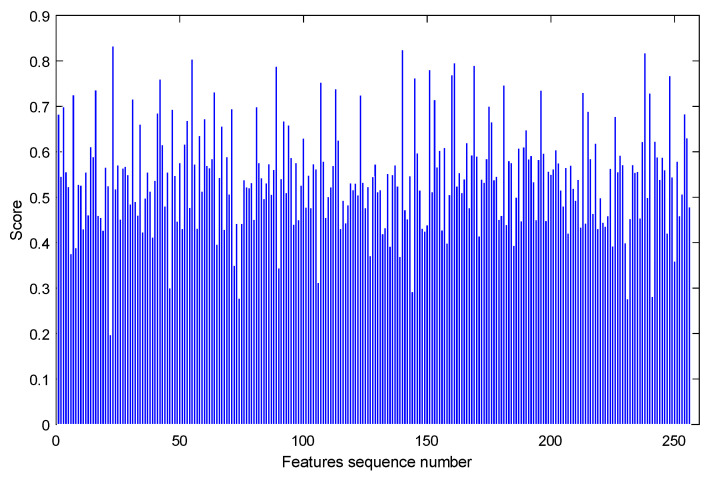
Trend consistency score of full feature sequence.

**Figure 13 sensors-23-08254-f013:**
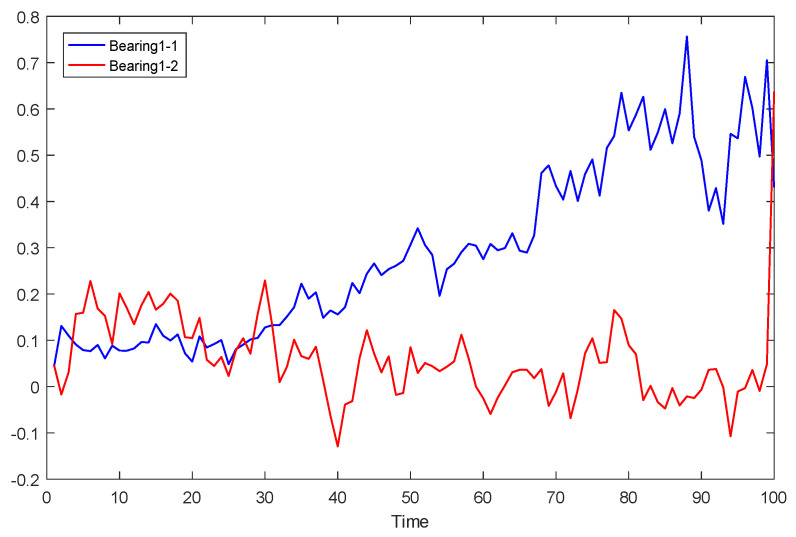
Trend term of the 22nd characteristic sequence of bearings 1_1 and 2_1 after down-sampling.

**Figure 14 sensors-23-08254-f014:**
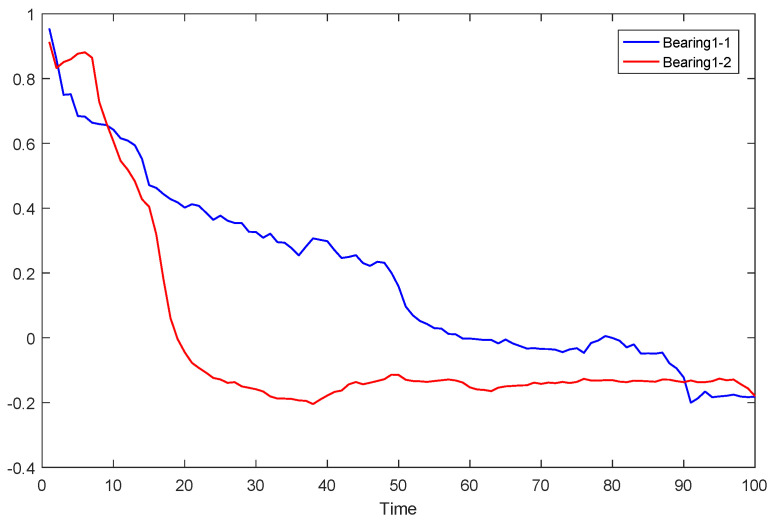
Trend term of the 23rd characteristic sequence of bearings 1_1 and 2_1 after down-sampling.

**Figure 15 sensors-23-08254-f015:**
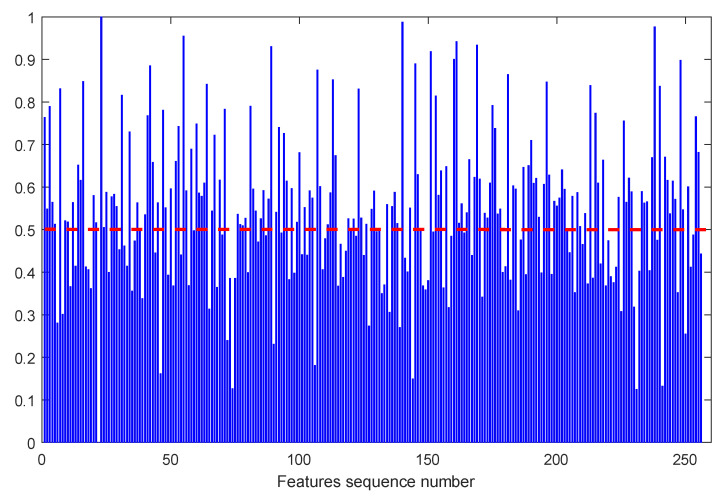
Normalized Score of Trend Consistency of Full Feature Sequence.

**Figure 16 sensors-23-08254-f016:**
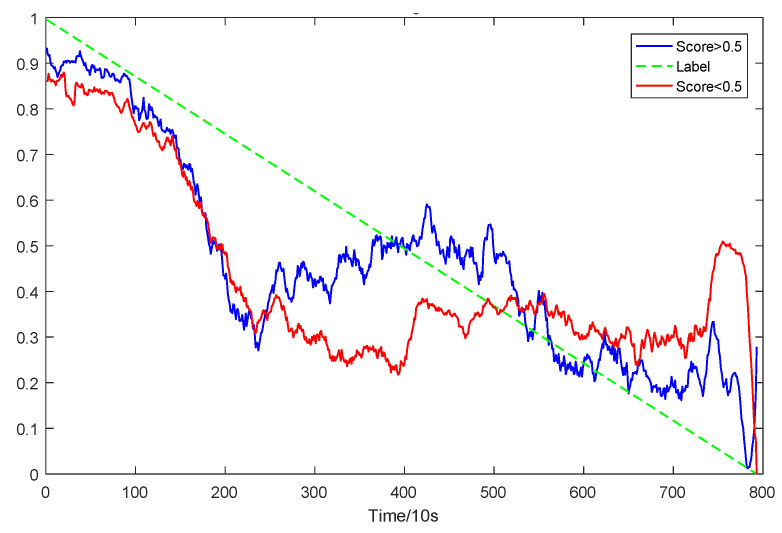
Prediction results of two kinds of feature sets of bearing 2_2.

**Figure 17 sensors-23-08254-f017:**
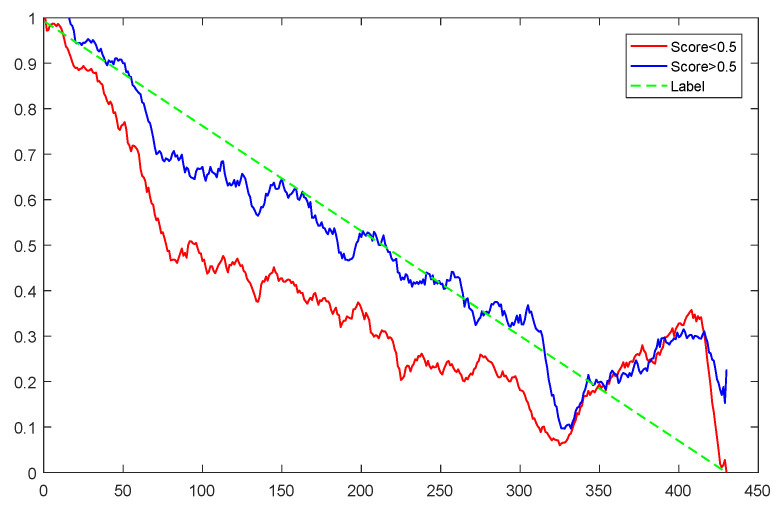
Prediction results of two types of feature sets of bearing 3_3.

**Table 1 sensors-23-08254-t001:** Autoencoder network model parameters.

Module		Layer	Kernel	Stride	Activation Function	Output Dimension
Feature extraction module	Encoder	Input layer	\	\	\	1 × 1280 × 1 × 1
		Convolution 1	20 × 1 × 64	8 × 1	Leaky_relu	1 × 160 × 1 × 64
		Convolution 2	9 × 1 × 128	4 × 1	Leaky_relu	1 × 40 × 1 × 128
		Convolution 3	9 × 1 × 256	4 × 1	Leaky_relu	1 × 10 × 1 × 256
		Flatten	\	\	\	1 × 1 × 1 × 256
	Decoder	Deconvolution 1	9 × 1 × 128	4 × 1	Leaky_relu	1 × 40 × 1 × 128
		Deconvolution 2	9 × 1 × 64	4 × 1	Leaky_relu	1 × 160 × 1 × 64
		Deconvolution 3	20 × 1 × 1	8 × 1	Leaky_relu	1 × 1280 × 1 × 1
		Output layer	\	\	\	1 × 1280 × 1 × 1

**Table 2 sensors-23-08254-t002:** Bearing data set.

**Load (N)**	4000	4200	5000
**Speed (rpm)**	1800	1650	1500
**Case**	Bearing Dataset1	Bearing Dataset2	Bearing Dataset3
	Bearing 1_1 Bearing 1_2 Bearing 1_3 Bearing 1_4 Bearing 1_5 Bearing 1_6 Bearing 1_7	Bearing 2_1 Bearing 2_2 Bearing 2_3 Bearing 2_4 Bearing 2_5 Bearing 2_6 Bearing 2_7	Bearing 3_1 Bearing 3_2

**Table 3 sensors-23-08254-t003:** Correlation value of the same feature.

	1-1	1-2	1-3	1-4	1-5	1-6	1-7	2-1	2-3	2-4	2-5	2-6	2-7
1-1	1	0.934	0.855	0.825	0.896	0.936	0.935	0.711	0.828	0.891	0.051	0.935	0.763
1-2	0.934	1	0.892	0.895	0.789	0.876	0.827	0.599	0.78	0.838	0.104	0.837	0.817
1-3	0.855	0.892	1	0.918	0.67	0.807	0.761	0.402	0.89	0.792	0.152	0.75	0.921
1-4	0.825	0.895	0.918	1	0.637	0.772	0.705	0.407	0.756	0.772	0.137	0.717	0.855
1-5	0.896	0.789	0.67	0.637	1	0.929	0.916	0.871	0.769	0.813	0.024	0.95	0.554
1-6	0.936	0.876	0.807	0.772	0.929	1	0.894	0.799	0.842	0.847	0.029	0.933	0.716
1-7	0.935	0.827	0.761	0.705	0.916	0.894	1	0.697	0.826	0.902	0.079	0.96	0.69
2-1	0.711	0.599	0.402	0.407	0.871	0.799	0.697	1	0.568	0.559	0.278	0.774	0.328
2-3	0.828	0.78	0.89	0.756	0.769	0.842	0.826	0.568	1	0.744	0.002	0.829	0.844
2-4	0.891	0.838	0.792	0.772	0.813	0.847	0.902	0.559	0.744	1	0.242	0.876	0.687
2-5	0.051	0.104	0.152	0.137	0.024	0.029	0.079	0.278	0.002	0.242	1	0.02	0.147
2-6	0.935	0.837	0.75	0.717	0.95	0.933	0.96	0.774	0.829	0.876	0.02	1	0.68
2-7	0.763	0.817	0.921	0.855	0.554	0.716	0.69	0.328	0.844	0.687	0.147	0.68	1

**Table 4 sensors-23-08254-t004:** The predicted results of the experiment.

Case	Full Feature Sequence	Time Correlation	Monotonicity	Robustness	Trend Consistency
2_1	0.217	0.228	0.238	0.227	**0.194**
2_2	0.127	0.126	**0.124**	0.132	0.128
2_3	0.208	0.196	0.200	0.201	**0.188**
2_4	0.26	0.265	0.271	0.268	**0.177**
2_5	0.262	0.274	0.271	0.275	**0.251**
2_6	0.257	0.241	0.229	0.207	**0.156**
2_7	0.304	0.335	0.297	0.296	**0.253**
3_1	0.189	0.186	0.183	0.191	**0.147**
3_2	0.175	0.198	0.198	0.209	**0.170**
3_3	0.180	0.146	0.165	0.24	**0.069**
Composite mean	0.2179	0.2195	0.2176	0.2246	**0.173**
Error reduction ratio	20.5%	21%	20.3%	22.8%	/

**Table 5 sensors-23-08254-t005:** Prediction experimental results of bearing 2-2 and bearing 3-3.

Index	Bearing 2_2	Bearing 3_3
Full feature sequence	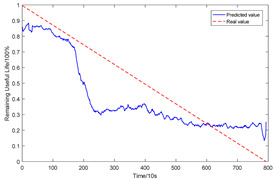	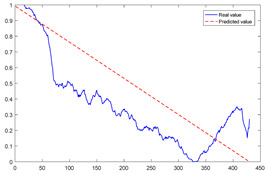
Time correlation	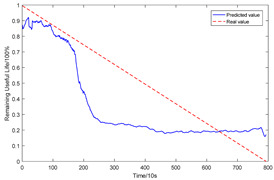	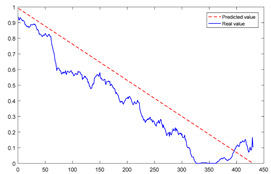
Monotonicity	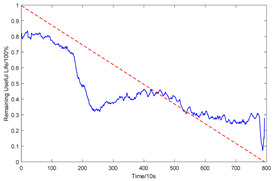	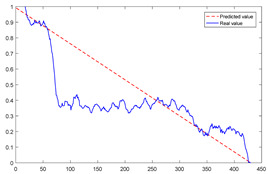
Robustness	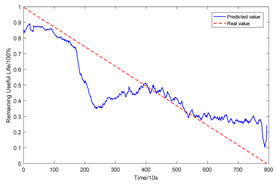	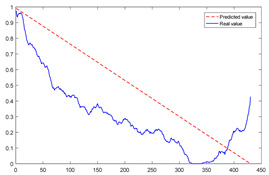
Trend consistency	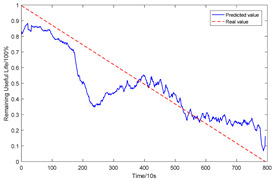	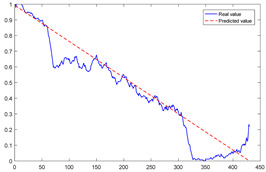

## Data Availability

The experimental data are vibration acceleration data collected from the accelerated life bench test of rolling bearing, which comes from the PHM data challenge [31] held by the Institute of Electrical and Electronics Engineers (IEEE) in 2012.

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
