# Peer review of "Transfer Prediction Method of Bearing Remaining Useful Life Based on Deep Feature Evaluation under Different Working Conditions"

_sensors, 2023, doi:10.3390/s23198254_

Round 1

Reviewer 1 Report

In this manuscript, a domain adversarial transfer model based on feature condition mapping is proposed to overcome the second limitation.  Authors was proposing a bearing RUL prediction method based on feature evaluation and deep transfer learning. The framework of the method includes feature extraction and evaluation, prediction, and domain adaptation module. Experimental results show that this method is superior to the existing bearing evaluation and prediction methods.

Author Response

Dear  Reviewers,

We would like to thank the reviewers for your constructive comments. We have revised the manuscript in accordance with your comments. All changes are shown in red in the manuscript.

We would like to express our great appreciation to  reviewers for comments on our paper. Looking forward to hearing from you.

Thank you and best regards.

Yours sincerely,

Yongzhi Liu

Corresponding author: Yongzhi Liu

Reviewer 2 Report

1.In the 2.4. Feature evaluation module, Step 2 refers to formula (3), in this article, the formula (3) is the Wasserstein distance between source domain feature distribution and target domain feature distribution, not the correlation formula.

2. In the 3.2. Feature extraction, why do the bearing2_3 get 320 feature sequences?

3.In the 3.4 Calculation and analysis of trend consistency index in feature evaluation, how was the 256 feature sequences obtained?

4.The captions of Figures 13 and 14 both show the trend term for the 23rd feature sequence of bearings 1-1 and 2-1 after downsampling.

5.In the conclusion (1) , it is mentioned that the overall prediction accuracy improved by 3.1%, but the method of obtaining this calculation result was not clearly explained in the text.

6. To further demonstrate the effectiveness and superiority of the method proposed in this paper, the author needs to supplement and compare it with other commonly recognized methods for bearing RUL prediction.

Author Response

(The authors gave the same response as above.)

Reviewer 3 Report

Dear Authors, thank you for your work. Problem of bearing monitoring is very important. Data collection and labeling for these algorithms are complicated, including transfer from lab or local tests to real setup/device. In the paper, you suggest the way in which data from limited number of tests can be expanded to more examples and I think it is interesting. I think it can be published after adding few comments for better understanding of chosen methods..   

References (need [x]) and figure/table links (showed errors) in the text are incorrect, please check.

Can you explain the choice of actication function in autoencoder, please?

Line 177 – need JS full name.

Eq.4 – describe ⊙ symbol.

Eq.5 - describe

Eqs. 6, 7 must looks like ‘X=Y+Z’, please rewrite.

Line 231 – wrong dot.

Eqs. 9-11 – I suggest to write them in contrary order and check that all symbols were described.

Line 257 – looks that correlation formula was lost, isn’t it?

Fig.9 – Can you make equal axis for both figures? Also please comment, haven’t you normalized features to 0..1 range?

Figure 13 – I guess 22nd characteristic in subscription.

Normalization of figure 12 to figure 15 was done with linear interpolation. Have you tried some nonlinear equation or thresholds <>0.5?

In figures 16-17 and table 5 you show linear decreasing as an ideal labeling. But as I understand in bearing work some faults can occur nonlinear in time (e.g. Fig 11). Can you please explain why you have chosen linear?

Also please subscribe all axes on figures and increase font size when it is too small for reading.

Author Response

(The authors gave the same response as above.)

Round 2

Reviewer 2 Report

The author has responded or made revisions to the problems raised in the first round of review one by one.

Author Response

Dear  Reviewers,

We would like to thank the reviewers for your  comments.

Thank you and best regards.

Yours sincerely,

Yongzhi Liu

Corresponding author: Yongzhi Liu

Reviewer 3 Report

Dear Authors, 

Unfortunately looks that some my comments were lost (though I see them on info pannel):

References (need [x]) and figure/table links (showed errors) in the text are incorrect, please check. (in my version all links to figures are errors, please check with Editors if it is some fail of my internet/PC/etc.)

In figures 16-17 and table 5 you show linear decreasing as an ideal labeling. But as I understand in bearing work some faults can occur nonlinear in time (e.g. Fig 11). Can you please explain why you have chosen linear?

Also please subscribe all axes on figures and increase font size when it is too small for reading.

Author Response

Dear  Reviewers,

We would like to thank  the reviewers for your constructive comments. We have revised the manuscript in accordance with your comments. All changes are shown in red in the manuscript. Here are the brief responses to your comments.

1- Unfortunately looks that some my comments were lost (though I see them on info pannel): References (need [x]) and figure/table links (showed errors) in the text are incorrect, please check. (in my version all links to figures are errors, please check with Editors if it is some fail of my internet/PC/etc.)

Answer: Thanks to your comments. It has been rechecked and corrected to the revised manuscript.

2- In figures 16-17 and table 5 you show linear decreasing as an ideal labeling. But as I understand in bearing work some faults can occur nonlinear in time (e.g. Fig 11). Can you please explain why you have chosen linear?

Answer: Currently, in the research on bearing life prediction, data-driven methods are commonly used. In this process, labeling the vibration signals is indeed a key step, but due to the nonlinear process of bearing damage, there is currently no unified theory or method to develop standard labels for vibration signals at each life stage. In other words, there is no standard definition for the lifetime label of a vibration signal, such as 0.5 or 0.7, when faced with a vibration signal.

To solve this problem, many studies have adopted a method of segmented labeling. That is, the vibration data of a full life cycle of the bearing is first divided into several segments, and then according to the characteristics of these segments, they are labeled proportionally within the full life cycle of 0-1. Although this method cannot completely and accurately predict the life of the bearing, it can improve the accuracy of prediction to some extent..

3- Also please subscribe all axes on figures and increase font size when it is too small for reading.

Answer: Thank you for your advice. Due to the large number of images and layout restrictions in the manuscript, the image size has been reduced, resulting in smaller font size for the displayed images. However, I have provided the original vector image for each image, and the text on the image can be clearly displayed when the vector image is enlarged. Next, I will make the layout according to the editor's requirements.

We would like to express our great appreciation to  reviewers for comments on our paper. Looking forward to hearing from you.

Thank you and best regards.

Yours sincerely,

Yongzhi Liu

Corresponding author: Yongzhi Liu
